# On the Testing of Advanced Automotive Radar Sensors by Means of Target Simulators

**DOI:** 10.3390/s20092714

**Published:** 2020-05-09

**Authors:** Premysl Hudec, Viktor Adler

**Affiliations:** Department of Electromagnetic Field, Faculty of Electrical Engineering, Czech Technical University in Prague, Technicka 2, 166 27 Prague 6, Czech Republic; hudecp@fel.cvut.cz

**Keywords:** automotive radar sensor, radar target simulator, sensor testing, sensor simulator setup, additive RF noise, phase noise

## Abstract

The rapid development and wide commercial implementation of automotive radar sensors are strengthening the already considerable interest in matching radar target simulators. Such simulators boast promising results when used for both essential functional inspections of active sensors and the high-speed testing of numerous traffic scenarios while examining complex reactions of automobile electronic systems. For these purposes, advanced versions of target simulators enabling a generation of multiple targets moving at different velocities and ranges are required. The design, practical implementation and system programming of advanced sensor simulator setups require a detailed analytical description concerning all important technical aspects. An abundance of detailed information on the behavior and parameters of automotive radar sensors can be found in the references, but similar knowledge on sensor simulator setups is lacking. This article presents detailed analyses of the all-important RF parameters, where special attention is paid to phase noise, and its analytical description takes into account an even greater number of simulated targets. The derived analytical formulas enable both an optimal setup implementation and system programming of a wide range of practical testing procedures.

## 1. Introduction

Automotive radars (ARs) can be rated as sensors with the most intensive development and expanding applications. At a minimum, three AR units, used as adaptive cruise control or blind spot sensors, are installed in almost every new vehicle, and in the near future, numerous sensors will be used to ensure 360-degree coverage [1]. The main task of all these sensors is to monitor a vehicle’s surroundings [2], i.e., utilize perception algorithms to detect and evaluate targets, such as neighboring vehicles, pedestrians, traffic signs and guardrails, among others. Based on this information, the electronic control unit (ECU) adapts the vehicle settings and initiates a suitable response, all of which should result in substantially improved road traffic safety [2]. In addition, due to their immunity to rain, dust or darkness, AR sensors will have an irreplaceable role in developed autonomous driving concepts. An overview of nowadays utilized modulation schemes, target classification and detection algorithms’ and inter-radars’ immunity can be found in [3]. As all of these electronics relate to the safety of people, all concerned sensors must be subject to intensive testing. For these purposes, target simulators are of great benefit.

A radar target simulator (TS) is a “box” placed in front of a tested radar that is capable of “convincing” a radar that it is detecting a target or multiple targets in required azimuths and ranges, and of showing the prerequisite velocities and appropriate radar cross-sections (RCSs). Such target simulators are used to check radar parameters or verify their calibration. In the automotive industry, radar target simulators can play an even more significant role. In practice, automotive electronic systems equipped with AR sensors must solve a virtually infinite number of traffic scenarios, resulting in extremely difficult, lengthy and costly test drives. This process becomes additionally cumbersome following any change to a car’s hardware (HW) or controlling software (SW) when all traffic scenarios must be tested again. TSs are also essential components in hardware-in-the-loop (HIL) testing and overall reliability evaluations.

Common AR sensors operate in the 76–77 GHz millimeter-wave frequency band, use frequency-modulated continuous-wave (FMCW) techniques, and measure ranges (*R*), radial velocities (*v*_r_) and azimuths (Ψ) of targets in their view [4,5]. Detailed information on these sensors including realization in a single-chip technology can be found in [6]. In recent years, a significant effort has been made to develop advanced automotive sensors. Beyond ensuring higher levels of immunity against mutual interferences by using more complex modulations, the development is, above all, focused on reaching a substantially higher accuracy and resolution [7,8]. Both goals can be met using a wider modulation frequency band in the 77–81 GHz region, and more transmitting (TX) and receiving (RX) channels. As a result, modern AR sensors detect significantly more targets than common types. For example, as described in [2], common narrow-bandwidth, low-resolution sensors detect a nearby vehicle as a single target. However, the surface of any vehicle consists of areas with low reflectivity and areas with high reflectivity. That is why advanced, wide-band, high-resolution automotive sensors “see” any neighboring car in its view as a set of many targets, each corresponding to one high-reflecting area [7,9]. This is true for all other type of targets.

There exist several wider families of TSs utilizing different principles and providing more of less advanced capabilities. Reference [10] describes a TS based on a reflector mechanically moved alongside a dielectric waveguide. In [11,12], a very simple TS structure without conversion is presented but its application is limited to FMCW radars only. The special TS used for the simulation of moving pedestrians and presented in [13] uses advanced down- and up-conversion circuits but enables only a Doppler frequency modulation. Solutions according to [14,15] employ switched optical fiber delay lines. In [16], a TS utilizing purely digital processing of digitalized radar signals is described, whereas a solution according to [17] depicts a system equipped with both digital and analogue processing branches. The former branch is supposed to ensure the simulation of far targets, while the latter is able to simulate nearby targets. Reference [18] describes a flexible TS solution based on commercial off-the-shelf measurement equipment.

According to Figure 1, a common TS typically consists of one ANT1 receiving antenna, target simulator circuits (TSC), i.e., all circuits between the TS receiving and transmitting antennas, and one ANT2 transmitting antenna. The receiving and transmitting antennas are situated in front of the tested sensor with a suitable distance for *R*_S_, ranging from 0.5 to 1 m. In TSCs, the signal from the tested sensor is received by the receiving antenna, delayed in time, shifted in frequency and amplified/attenuated. The time delay corresponds to the required range of the simulated target, and the frequency shift simulates the Doppler frequency shift resulting from the target´s radial velocity, while the amplitude of the transmitted signal corresponds to the target´s RCS and range. The processed signal is then sent back to the sensor. Such a sensor simulator setup can simulate a single target (or several targets at different ranges) at a single azimuth defined by the sensor’s TS transmitting antenna’s interconnecting line. Simulated target motions can occur at this line only. However, to simulate real traffic scenarios, it is necessary to generate a higher number of targets at more azimuths while the velocity vectors can also evince the azimuthal components (components in directions perpendicular to the sensor’s TX antenna’s interconnecting lines).

As one approach to these requirements, the setup described in [19] employs a single TS receiving antenna fixed in front of the tested AR and five TS transmitting antennas which can be mechanically moved alongside sections of circles in view of the tested AR sensor. This structure makes it possible to simulate up to five independent targets at five different azimuths with independent radial and azimuthal velocity components. This represents an advanced solution, but it is still far from the “raw” radar pictures “seen” by advanced AR sensors in real traffic.

References [20,21] propose an interesting solution of how to generate really complex radar pictures. This solution is based on an array of TS transmitting antennas, which can be seen in Figure 2, where each TS transmitting antenna is connected to a TXM*_i_* independent signal processing and transmitting module. The high number of installed array elements enables a high number of targets at arbitrary ranges and azimuths to be generated in parallel. Switching targets by ΔΨ from one azimuth to another simulates motions with the azimuthal velocity components. Despite having been published in 1987 as a tool for testing missile radars, up to now, this initial design and the realization of this type of testing setup have not yet matured. To accelerate its development, it is, above all, necessary to derive an acceptably simple and economically acceptable HW structure. This primarily concerns the most demanding and costly millimeter-wave and digital circuits. The following sections indicate that the required simplification can be derived from a detailed analysis of the sensor simulator setup in question. The same analysis is also beneficial for system programming, that means specific settings of all system circuits and parameters, which can enable the generation of the required complex radar pictures. Considering AR sensors alone, their structures, parameters, algorithms, applications, etc., can be found in a great number of references, some of which are mentioned within this text. This also concerns a detailed analysis of additive RF noise and the phase noise of FMCW sensors which can be found even in somewhat dated references [22,23]. Unfortunately, a detailed theoretical analysis of the AR sensor–target simulator setup under concern is lacking. This article aims to fill this gap.

The article presents both derived analytical formulas describing all essential system parameters and recommendations on how they contribute to the optimization of the HW setup structure and to system programming. Section 2 starts with a signal level analysis, while Section 3 is devoted to an additive RF noise analysis. As local oscillators (LOs) are inevitable parts of both advanced sensors and simulators, Section 4 is dedicated to related phase noise problems. Section 5 studies the same phenomenon in relation to the generation of multiple targets. Results of the calculated parameters and performed computer simulations are summarized in Section 6. Section 7 evaluates the obtained results with respect to the optimal HW structure and system programming.

## 2. Signal Level Analysis

Figure 1 depicts fundamental AR sensor–TS relations. The AR transmitter transmits output power (*P*_T_) with its output antenna gain equal to *G*_RT_. The TS receiving antenna with gain (*G*_SR_) and TS transmitting antenna with gain (*G*_ST_) are situated at distances (*R*_S_) from the AR sensor. The gain (*G*_S_) depicts the overall TSC system gain, excluding the gains of the input and output antennas, while *G*_RT_ stands for gain of the sensor’s receiving antenna.

Using the above-defined variables, the AR input power (*P*_R_) obtained through the TS can be expressed as
(1)$$P_{R}=P_{T}\frac{G_{\mathrm{RT}}G_{\mathrm{SR}}G_{S}G_{\mathrm{ST}}G_{\mathrm{RR}}\lambda^{4}}{R_{S}^{4}{\left(4\pi \right)}^{4}}$$
The power (*P*_R_) can be compared with power which corresponds to the AR detecting a real target (*P*_Rt_) with the RCS (σ) situated at a range (*R*_t_), see Figure 3. This power can be evaluated using the standard radar equation
(2)$$P_{\mathrm{Rt}}=P_{T}\frac{G_{\mathrm{RT}}G_{\mathrm{RR}}\sigma \lambda^{2}}{{\left(4\pi \right)}^{3}R_{t}^{4}}$$
Comparing Equation (1) and (2), the RCS of the simulated target or, for design purposes, the overall TSC system gain (*G*_S_) can be derived as
(3)$$G_{S}=\sigma \frac{4\pi R_{S}^{4}}{G_{\mathrm{SR}}G_{\mathrm{ST}}\lambda^{2}R_{t}^{4}}$$
As mentioned, *G*_S_ corresponds to the gain of the overall TS circuits which provide a constant RCS of the simulated target. This gain includes the gains of all millimeter-wave circuits, intermediate frequency (IF) circuits, base-band (BB) circuits and, with care, the digital circuits used and the signal processing applied. It is one of the most important parameters for TS design. When designing TS output circuitry, the output TS power (*P*_ST_) can be useful:(4)$$P_{\mathrm{ST}}=P_{\mathrm{SR}}G_{S}$$
where *P*_SR_ is the power received by the AR. In specific programming steps described more closely in Section 7, the solution of Equations (1) and (2), with respect to the RCS, can be necessary, and the resulting formula can be expressed as
(5)$$\sigma =\frac{G_{S}G_{\mathrm{SR}}G_{\mathrm{ST}}\lambda^{2}R_{t}^{4}}{4\pi R_{S}^{4}}$$

Using this formula, an effective RCS of the simulated target can be calculated as a function of the currently set system parameters. Examples of some practically calculated values based on formulas derived in this section are presented in Section 6.

## 3. Additive RF Noise Analysis

As with any radio system, the operation of the TS is influenced by both additive RF noise and phase noise. This section considers the former, but firstly, the signal-to-noise ratio (*SNR*_Rt_) corresponding to a real target must also be evaluated, see Figure 3. Consequently, it must be determined how the TS influences the AR sensor noise behavior. The *SNR*_Rt_ is related to the input of the AR sensor receiver and can be expressed using the formula
(6)$$SNR_{\mathrm{Rt}}=\frac{P_{\mathrm{Rt}}}{kT_{0}B+N_{\mathrm{aR}}}$$

In this equation, *k* = 1.38×10^−23^ J/K represents the Boltzmann constant, while *B* equals the noise bandwidth. This bandwidth depends upon the sensor type and signal processing applied, e.g., during the evaluation of range in common FMCW sensors, it equals a width of one FFT (Fast Fourier Transform) bin. In Equation (6), *P*_Rt_ stands for the AR input signal power received through a reflection from a real target and this power can be evaluated using Equation (2). *T*_0_ = 290 K is a standard thermodynamic temperature. Noise power (*N*_aR_) added by the AR, and related to the input of its receiver, is a function of the AR sensor noise figure *(F*_R_), and is defined by the formula
(7)$$N_{\mathrm{aR}}=\left(F_{R}-1\right)kT_{0}B$$

The sensor–TS structure represents a cascade of RF components with two free-space propagation paths showing high attenuations. Since the surrounding temperature of all involved antennas should be close to *T*_0_, for the purposes of the RF noise analysis, these signal paths can be modeled using matched attenuators showing overall attenuations *L*_1_ and *L*_2_, see Figure 1. Considering the AR sensor–TS signal path, *L*_1_ attenuation can be expressed as
(8)$$L_{1}=\frac{{\left(4\pi R_{S}\right)}^{2}}{\lambda^{2}G_{\mathrm{RT}}G_{\mathrm{SR}}}$$

The *L*_2_ attenuator models the TS–AR sensor signal path and its value can be calculated using the following relation:(9)$$L_{2}=\frac{{\left(4\pi R_{S}\right)}^{2}}{\lambda^{2}G_{\mathrm{ST}}G_{\mathrm{RR}}}$$

According to [24], available noise powers generated by both attenuators *L*_1_ and *L*_2_ equal *kT*_0_*B*, and this relation describes a logical phenomenon that, from a noise generation point of view, any well-matched attenuator with high loss will behave like a matched load. The noise figure of the TSC equals *F*_S_, while its gain (*G*_S_) is described by Equation (3). Noise added by the TS related to its input can be evaluated as
(10)$$N_{\mathrm{aS}}=\left(F_{S}-1\right)kT_{0}B$$

All noise contributions can be summed in the A-A’ plane at the input of the AR sensor receiver, and the corresponding noise floor is
(11)$$N_{\mathrm{RS}}=\left(kT_{0}B+N_{\mathrm{aS}}\right)\frac{G_{S}}{L_{2}}+kT_{0}B+N_{\mathrm{aR}}$$

The resulting signal-to-noise ratio (*SNR*_RS_) related to the AR sensor input, and corresponding to an artificial target simulated by the TS, can be evaluated using
(12)$$SNR_{\mathrm{RS}}=\frac{P_{\mathrm{Rt}}}{N_{\mathrm{RS}}}$$

For TS design, it may be useful to know what the highest TSC noise figure *(F*_Smax_) is, or the added noise power (*N*_aSmax_) which degrades the *SNR*_RS_, with respect to the *SNR*_Rt_, by a still acceptable degree. This can be evaluated using the relation
(13)$$SNR_{\mathrm{RS}}=K_{\mathrm{RF}}\cdot SNR_{\mathrm{Rt}}$$

In this condition, *K*_RF_ stands for the coefficient of the allowed *SNR*_Rt_ drop by additive RF noise, e.g., K_RF_ = 0.5 for the –3 dB drop considered. If the TS shows the same RCS as the real target, hence *P*_RS_ = *P*_Rt_, the maximum possible noise (*N*_aSmax_) added by the TS, and related to the input of TSC, can be expressed using Equations (12) and (6) as
(14)$$N_{\mathrm{aSmax}}=\frac{\left(kT_{0}B+N_{\mathrm{aR}}\right)\left(1-K_{\mathrm{RF}}\right)L_{2}}{K_{\mathrm{RF}}G_{S}}$$
From Equation (10), the maximum (*F*_Smax_) TSC noise figure can be evaluated as
(15)$$F_{\mathrm{Smax}}=\frac{N_{\mathrm{aSmax}}}{kT_{0}B}+1$$
or, using Equations (3), (7) and (14) as
(16)$$F_{\mathrm{Smax}}=\frac{F_{R}L_{2}G_{\mathrm{SR}}G_{\mathrm{ST}}\lambda^{2}R_{t}^{4}\left(1-K_{\mathrm{RF}}\right)}{4\pi R_{S}^{4}K_{\mathrm{RF}}\sigma }+1$$

It is clear that for an ideal case where the TS circuitry does not worsen the *SNR*_RS_ (*K*_RF_ = 1), the highest TSC noise figure equals *F*_Smax_ = 1, but quickly increases when even a slight (tenths of dB) *SNR*_RS_ degradation is allowed.

Examples of practical values are presented in Section 6. The calculated values of *F*_Smax_ are perhaps even surprisingly higher than can be expected from the low range of *R*_S_. The high obtained values of *F*_Smax_ are naturally advantageous and they can lead to simpler than usual millimeter-wave down-converting structures. On the other hand, the TS cannot be fully passive, and amplifiers must be used at suitable system positions. That is why their noise figures must be considered and the noise figure of the whole TSC carefully evaluated.

## 4. Phase Noise Influence

References [10,11,12] are probably the only ones depicting automotive simulators working directly in the AR sensor operational frequency band. The solution in [10] works in the 76–77 GHz band and is based on a long section of a dielectric waveguide equipped with a fast sliding reflecting ring. Its operating principle is simple and it does not require any complex circuitry. Nonetheless, it shows great dimension, while enabling only a single target to be simulated, with its range limited to 20 m and the radial velocity a mere ± 30 km/h. The usage of simulators, according to [11], is limited to FMCW radars, described, e.g., in [5] and [23]. The application of this solution is also troubled by the fact that many difficult to ascertain sensor parameters must be known in advance. All the more versatile and efficient solutions are based on frequency conversion and signal processing performed at substantially lower intermediate frequencies (IF) or base-band (BB) frequencies, see, e.g., [13,14,15,16,17,18].

Figure 4 depicts a TS structure using standard millimeter-wave down-converting and up-converting circuits. The signal received by the ANT1 input antenna is amplified by a low-noise amplifier (LNA) and, using the MIX1 millimeter-wave mixer and the LO1 local oscillator, converted from the AR sensor operational frequency band *f*_0_ ± *B*_R_/2 into the intermediate frequency band *f*_IF_ ± *B*_R_/2. The frequency band *B*_R_ equals the AR sensor’s modulation bandwidth. In the IF frequency band, all signal processing steps necessary for the target simulator operation are performed. Consequently, the processed signal is converted back to the AR sensor frequency band using the up-converter consisting of the MIX2 millimeter-wave mixer and the LO1 local oscillator. Since the same LO1 is used for both the down- and up-conversions, the process is coherent and precise output frequencies can be reached. The power amplifier PA is needed if an output power (*P*_ST_) exceeding approximately −10 dBm is required, which is nearly always true in the communication field.

Frequency conversions enable efficient signal processing but bring a significant potential problem in that a phase noise is introduced into both sensors and simulators from the employed LOs. For both sensors and simulators, one LO is used for both the down- and up-conversions, which is why the concerned phase noises can be mutually correlated and time delay-dependent.

Considering practically used AR sensors, it is generally very difficult to make any assumptions on their phase noise [3]. When evaluating the concerned phase noise influences, it is necessary to know many important parameters and a great deal of information. In addition to the HW parameters, such as the AR sensor output power, the power spectrum density (PSD) of its LO phase noise, cross-talk gain or time delay of this cross-talk, it is necessary to know the modulation scheme and applied signal processing methods. Unfortunately, AR sensor manufacturers only rarely provide such information and it is not usually possible to measure them in a reasonable way. That is why, to perform a basic estimate of the influence of the simulator’s inner LO on all sensor simulators’ noise behavior, the influence of phase noise on the sensor’s inner LO is considered to be negligibly low.

The simplified analysis considers the TS with a single down-converting and up-converting chain and ideal delay block, as can be seen in Figure 5a. In this scheme, the same LO is used for both the down- and up-conversion. In the time domain, phase noise can be described using random phase fluctuations (*φ*(*t*)) which add to the general phase of the LO signal. From a frequency point of view, phase noise represents random frequency fluctuations (Δ*f*). In the case of a zero time delay (*τ*_mem_) (practically unachievable) of the TS, the resulting phase noise influences equal zero. The concerned frequency fluctuations are eliminated in the down- and immediate up-conversions, generally described as
(17)$$f_{\mathrm{RF}}-\left(f_{\mathrm{LO}}+\Delta f\right)+\left(f_{\mathrm{LO}}+\Delta f\right)=f_{\mathrm{RF}}$$

Considering a real TS, the delay (*τ*_mem_) consists of a latency of both the analog and digital TS circuits, and an intentionally set delay simulating a round-trip of the radar signal to the target and back. The utilized LO signal includes a phase noise described by random phase fluctuations (*φ*(*t*)). Due to the multiplicative nature of phase noise, i.e., phase noise power is always directly proportional to a power of the deterministic signal, the following analysis neglects signal amplitudes. Hence, the complex LO signal can be expressed as
(18)$$s_{\mathrm{LO}}\left(t\right)=\exp\left(j\left(2\pi f_{\mathrm{LO}}t+\varphi \left(t\right)\right)\right)$$

Since phase noise of the AR is supposed to be negligible, the received RF signal can be described as a pure harmonic signal by
(19)$$s_{\mathrm{RX}}\left(t\right)=\exp\left(j\left(2\pi f_{\mathrm{RF}}t\right)\right)$$

The down-conversion process performed in the TS can be described as a multiplication of the received and LO signals by
(20)$$s_{\mathrm{IF}}\left(t\right)=s_{\mathrm{RX}}\left(t\right)s_{\mathrm{LO}}^{*}\left(t\right)=\exp\left(j\left(2\pi f_{\mathrm{IF}}t-\varphi \left(t\right)\right)\right)$$
where * represents the complex conjugate. After the introduction of the time delay (*τ*_mem_) and up-conversion, the signal (*s*_TX_) transmitted back from the TS to the AR sensor can be described as
(21)$$s_{\mathrm{TX}}\left(t\right)=s_{\mathrm{IF}}\left(t-\tau_{\mathrm{mem}}\right)s_{\mathrm{LO}}\left(t\right)=\exp\left(j\left(2\pi f_{\mathrm{RF}}t+\alpha_{0}\left(\tau_{\mathrm{mem}}\right)+\Delta \varphi \left(t,\tau_{\mathrm{mem}}\right)\right)\right)$$

In this equation, α_0_(*τ*_mem_) represents a general constant phase shift, while Δ*φ*(*t*_,_
*τ*_mem_) describes time-dependent random phase fluctuations:(22)$$\Delta \varphi \left(t,\tau_{\mathrm{mem}}\right)=\varphi \left(t\right)-\varphi \left(t-\tau_{\mathrm{mem}}\right)$$

Such a propagation of phase noise from the LO through the down-conversion, time delay and up-conversion described by Equation (21) can be modeled using the circuit shown in Figure 5b. The impulse response of this circuit can be expressed as
(23)$$h\left(t\right)=\delta \left(t\right)-\delta \left(t-\tau_{\mathrm{mem}}\right)$$
where δ represents a Dirac delta function and a time delay (*τ*_mem_) of the entire chain generating an artificial target at the required distance. A voltage transfer function (*H* (Δ*f*)) of the circuit equals the Fourier transform of Equation (23):(24)$$H\left(\Delta f\right)=\int_{-\infty }^{\infty }h\left(t\right)e^{-j2\pi \Delta ft}dt=1-e^{-j2\pi \Delta f\tau_{\mathrm{mem}}}$$

Into this equation, the following frequency transform was introduced:(25)$$\Delta f=f-f_{b}$$

By considering both the positive and negative values of Δ*f*, this transform enables the concerned phase noise frequency dependences in the vicinity of any beat frequency (*f*_b_) corresponding to the FMCW principles to be described. Based on Equation (24), the power transfer function can be expressed as
(26)$$G\left(\Delta f\right)={\left|H\left(\Delta f\right)\right|}^{2}=4\sin^{2}\left(\pi \Delta f\tau_{\mathrm{mem}}\right)$$

Frequency dependence of the transfer function *G*(Δ*f*) as a parametric function of the simulated target ranges and plotted in the dB scale for the positive values of Δ*f* in a logarithmic frequency scale can be seen in Figure 6.

Owing to strong noise correlation, very short simulated target ranges (short time delays) lead to a nearly ideal subtraction and suppression of phase noise at low frequency offsets. Generally, when simulating targets at higher ranges, time delay values are higher and phase noise at the up-converter output is less correlated, therefore, phase noise suppression is not only substantially lower, but in definite frequency ranges it can be “amplified” up to 6 dB, though it is not a real amplification as the positive transfer function values in Figure 6 reflect the positive interferences of noise in certain frequency bands.

The phase noise properties of practical RF oscillators are typically described by the function ℒ(f) as a PSD of the phase noise related to the carrier power and expressed for the positive frequencies of Δ*f* only [25]. It can be measured or found in a datasheet of the LO utilized. An example of the function ℒ(f) corresponding to a standard PLL-based mmW (milimeter-wave) oscillator can be seen in Figure 7.

Since one of the following equations requires the integration of PSD phase noise in an FFT window, unsymmetrical with respect to the carrier, the function ℒ(f) was mirror-extended to cover both the negative and positive frequencies of Δ*f*. In the following text, this DSB phase noise function is indicated as *S*_LO_(Δ*f*).

The resulting phase noise spectrum density (*S*_Δ*φ*_ (Δ*f*)) at the up-converter’s output can be evaluated as
(27)$$S_{\Delta \varphi }\left(\Delta f\right)=S_{\mathrm{LO}}\left(\Delta f\right)G\left(\Delta f\right)$$

An example of the values of the *S*_Δ*φ*_ (Δ *f* ) evaluated for the PLL phase noise specified in Figure 7 can be seen in Figure 8.

The impact of TS phase noise on an AR sensor’s overall performance depends on the radar modulation scheme and applied signal processing. For example, as mentioned earlier, FMCW radars evaluate target ranges from beat signal frequencies *(f*_b_). For this purpose, before being processed by the FFT, the IF or BB signals at the outputs of radar receivers are sampled by suitable ADCs throughout almost an entire duration of frequency chirps (*T*_sw_). The corresponding noise bandwidth *B* usually equals 1/*T*_sw_, and determines the resolution in a spectrum [26]. The phase noise power in the digitized IF or BB signal can be evaluated as
(28)$$N_{\mathrm{PN}}=P_{R}\int_{f_{1}}^{f_{2}}S_{\Delta \varphi }\left(f-f_{b}\right)df$$

In this equation, the frequency bounds correspond to the edges of individual FFT bins following *f*_2_ – *f*_1_ = *B*. For the target range evaluation, the bin containing the beat frequency (*f*_b_) is decisive; the influences of phase noise in other bins are unimportant.

Figure 8 indicates that any integral of Equation (28) of the *S*_Δ*φ*_ (Δ*f*) plot in a common *B* = 12.5 kHz-wide FFT window, including the beat frequency (*f*_b_) at Δ*f* = 0, leads to very low values of *N*_PN_, resulting in negligible effects of TS phase noise on the AR sensor’s overall noise behavior. This situation can significantly differ if a simulation of multiple targets is required.

## 5. Simulation of Multiple Targets

The generation of complex radar pictures may require the simulation of *M* > 1 targets (but typically *M* ≤ 3) at a single azimuth, i.e., at a single sensor’s TS transmitting antenna’ connecting line. Theoretically, only the nearest target should be detected, while the more distanced reflections should be shaded. In practice, automotive sensors often detect taller targets standing or moving behind the nearest target. In this case, the phase noise problems described in Section 4 can become more serious. This phenomenon results from the overlapping of the phase noise spectra around the beat signals corresponding to nearby targets.

According to, e.g., [6], the range (*R*_t*i*_) of the *i*-th target measured by the FMCW sensor can be evaluated from the detected *i*-th beat frequency (*f*_b*i*_) as
(29)$$R_{ti}=\frac{f_{bi}c_{0}T_{\mathrm{sw}}}{2B_{R}}$$
where *c*_0_ is the speed of light. In this equation, *B*_R_ represents the FMCW radar modulation bandwidth (chirp bandwidth), while *T*_sw_ stands for the chirp duration. The total phase noise power (*N*_PN*i*_) degrading the beat signal of the *i*-th simulated target takes into account the PSD of its own phase noise and the phase noise of *M*–1 neighboring targets simulated at the same sensor simulator’s TX connecting line. The phase noise power (*N*_PN*i*_) can be calculated as a summation of the phase noise powers of Equation (28) from of all the individual targets in a *B* spectrum frequency bin width as
(30)$$N_{\mathrm{PN}i}=\sum_{j=1}^{M}P_{Rj}\int_{f_{1i}}^{f_{2i}}S_{\Delta \varphi j}\left(f-f_{bj}\right)df$$

Considering the worst case *f*_b*i*_ positions, the maxima of the PSD of the phase noise of the neighboring targets fall at the beat frequency of the middle target. According to Figure 8, the PSD maxima occur around the frequency offset Δ*f* =*f*_bc_ = 100 kHz, and by using Equation (29), the critical distance (Δ*R*_tc_) between the simulated targets can be evaluated as
(31)$$\Delta R_{\mathrm{tc}}=\frac{f_{\mathrm{bc}}c_{0}T_{\mathrm{sw}}}{2B_{R}}$$

According to Equation (27), in addition to the transfer function *G*(Δ*f*), the maxima of the function *S*_Δ*φ*_(Δ*f*) strongly depend upon the phase noise properties of the microwave frequency source used. Modern LOs are based on the PLL structure which, within a bandwidth of its low-frequency loop, reduces the PSD of the phase noise around the carrier to an approximately constant value (${ℒ}_{P}$) shaped into a so-called pedestal ([26,27]). In an example presented in Figure 7, only the right end of this phase noise pedestal determines the maxima of the function *S*_Δ*φ*_(Δ*f*), and therefore the critical beat frequency (*f*_bc_) and the critical distance (Δ*R*_tc_) between the two targets, respectively.

The common values *T*_sw_ = 80 μs and *B*_R_ = 600 MHz used in Section 6 lead to Δ*R*_tc_ = 2 m which may be quite a realistic inter-target distance. In Figure 9, the simulated spectra of the IF or BB radar signal, with *M* = 3-detected targets with identical RCSs, are plotted. All considered targets share the same LO characterized by the PSD of its phase noise, as defined in Figure 7. The middle target is exactly 2 m from the outer targets, i.e., about 100 kHz on the frequency axis. The beat signal (*S*_IF2_) corresponding to the middle target at the distance *R*_t2_ = 15 m is degraded by a phase noise originating from both outer beat signals. Considering the approximately constant values of *S*_Δ*φi*_(Δ*f*) in a frequency band *B* around *f*_bc_, the worst case phase noise power (*N*_PNw_) in the single FFT bin containing the middle beat frequency can be estimated as
(32)$$N_{\mathrm{PNw}}=B\left(P_{R1}S_{\Delta \varphi 1\;}\left(f_{\mathrm{bc}}\right)+P_{R3}S_{\Delta \varphi 3}\left(f_{\mathrm{bc}}\right)\right)$$
where *P*_R1_ and *P*_R3_ depict the powers of beat signals of the two outer targets, while *S*_Δφ1_(*f*_bc_) and *S*_Δφ3_(*f*_bc_) represent the maximal values of the PSD of the correlated phase noise according to Equation (29). Due to the generally small value of *S*_Δφ_ for small frequency offsets (see Figure 8), the degradation of the beat signal of the middle target by its own phase noise is considered to be negligible. The above-described behavior is illustrated in Figure 9.

This graph depicts beat signals, together with the frequency dependences of the phase noise powers (*N_PN_*) integrated in the *B* = 12.5 kHz windows, and related to three targets simulated at the ranges equal to 13, 15 and 17 m. The phase noise powers related to the individual beats were calculated using Equation (28) and the phase noise power corresponding to the phase noise floor was calculated using Equation (30). It can be clearly seen that the worst case phase noise power level corresponds to the 15-m-target, but significant phase noise levels are apparent even around both side targets. If *R*_t1,3_ >> *R*_tc_, the received powers and the PSD of the correlated phase noise from both side targets are almost identical, and assuming an approximately constant value for ${ℒ}_{P}$, another degree of simplification can be derived:(33)$$N_{\mathrm{PNw}}=B\left(P_{R1}S_{\Delta \varphi 1\;}\left(f_{\mathrm{bc}}\right)+P_{R3}S_{\Delta \varphi 3}\left(f_{\mathrm{bc}}\right)\right)\approx 2BP_{R}S_{\Delta \varphi \;}\left(f_{\mathrm{bc}}\right)\approx 2BP_{R}G\left(f_{\mathrm{bc}}\right){ℒ}_{P}$$

From the resulting value of *N*_PNw_, corresponding to the 15-m-target, requirements placed on the target simulator LO phase noise qualities can be determined. Since no practical data on AR sensor phase noise are generally available, the influence of TS phase noise was related to the AR sensor’s additive RF noise properties, which can be estimated much more easily and much more precisely. As with the case of the additive RF noise analysis presented in Section 3, an allowed drop (*K*_PN_) of the *SNR*_Rt_ of the detected beat signal occurs at the AR sensor’s input and this can be evaluated using the relation
(34)$$SNR_{\mathrm{RP}}=K_{\mathrm{PN}}\cdot SNR_{\mathrm{Rt}}$$

In this formula, i.e., *K*_PN_ = 0.5 for a −3 dB drop, *K*_PN_ = 0.1 for a −10 dB drop, etc., the *SNR*_RP_ represents the signal-to-noise ratio of the power of the detected beat signal of the 15-m-target, with respect to the total influence of the additive RF noise power and worst case phase noise evaluated by Equation (30), and related to the input of the AR sensor’s receiver. From the FMCW sensor point of view, both additive RF noise and phase noise show similar behavior. Both noises are uncorrelated, increase the noise floor and vary the phase of the detected beat signals. That is why their total influence can be evaluated as the sum of the above stated noise powers:(35)$$SNR_{\mathrm{RP}}=\frac{P_{R}}{kT_{0}B+N_{\mathrm{aR}}+N_{\mathrm{PNw}}}$$

Using Equations (2), (7), (26), (33), (34) and (35), it is possible to express the boundary phase noise pedestal value (${ℒ}_{\mathrm{Pb}}$) of the LO source within the TS which *K*_PN_-times degrades the signal-to-noise ratio (*SNR*_Rt_) in a single *B*-wide FFT bin as
(36)$${ℒ}_{\mathrm{Pb}}=\frac{\left(kT_{0}B+N_{\mathrm{aR}}\right)\left(1-K_{\mathrm{PN}}\right){\left(4\pi \right)}^{3}R_{t}^{4}}{8K_{\mathrm{PN}}P_{T}G_{\mathrm{RT}}G_{\mathrm{RR}}\sigma \lambda^{2}B\sin^{2}\left(\pi f_{\mathrm{bc}}\frac{2\left(R_{t}-R_{S}\right)}{c_{0}}\right)}$$

It should be emphasized that this equation corresponds to a simplification of Equation (33), meaning that the evaluation of Equation (36) for near targets provides less accurate results.

The practical values of the allowed phase noise properties of a TS´s LO source, evaluated according to Equation (36) and presented in the next section, indicate that the influence of phase noise on a sensor simulator’s behavior is still relatively low and that even relatively “dirty” LOs can be acceptable. As stated in Section 7, this fact simplifies the design and construction of the concerned sensor simulator setups.

## 6. Application and Verification of Derived Formulas

This section presents practically calculated values corresponding to the TS suitable for the testing of FMCW medium-range radars (MRRs). As a reference unit, the Bosch MRR1 device operating in the 76–77 GHz band, and described in [6] and [28], is considered. Many of its parameters are not available to the public, however, they were estimated from an open sample of this AR sensor and spectrum-analyzer measurements. The obtained parameters necessary for the calculations presented in this section are summarized in Table 1.

Values presented in Table 1 were read from the performed AR measurements or estimated from an opened AR sample. Valuable information can also be found in the references, especially [6]. The measurements were performed using a W-band receiving horn antenna, mixer with a 75.9 GHz LO signal and oscilloscope with 5 GHz sampling. Figure 10 depicts the measured spectrogram showing the bandwidth of the frequency chirps and active sweep time. The transmitting antenna is formed by an 8 × 10 array and its gain was estimated to 20 dB, while the gain of the receiving 1 × 8 array was estimated to be 10 dB. The AR sensor receiver noise figure was assessed according to noise figures of similar 80 GHz LNAs, and significant reserves were added to cover the additional losses. The presented sampling frequency value is based on consultations with experts involved in the AR problems.

Let us suppose a TS equipped with two identical horn antennas with gains *G*_SR_ = *G*_ST_ = 14 dB and AR sensor–TS distance equal to *R*_S_ = 0.5 m. With respect to [6], all further calculations consider targets with an RCS *σ* = 1, 10 and 100 m^2^, and simulated target ranges of *R*_t_ = 3, 10, 30 and 100 m. These values should sufficiently cover the requirements for MRR testing involving pedestrians, cars, trucks, traffic signs or guardrails.

The RF power received by the TS reaches *P*_SR_ = –20.1 dBm behind the TS’s receiving antenna, while the calculated values of the TS system’s gain (*G*_S_) providing a constant RCS, as defined by Equation (3), are stated in Table 2. The values range from –60 dB to +20 dB which represents an 80 dB required dynamic range. In practice, this dynamic range can be significantly lower. Usually, AR sensors show relatively narrow antenna radiation patterns and large targets in near ranges are not fully illuminated. In addition, in near ranges, for proper target detection, it is not necessary to simulate targets with very high RCSs. Therefore, a *G*_smax_ not higher than 0 dB and a dynamic range lower than 60 dB can be considered as satisfactory. This dynamic range can be easily realized, for example, by a combination of digitally controlled analog attenuators.

The ability of the TS, featuring a passive millimeter-wave up-converter (Figure 4), to simulate targets at near ranges with a high RCS, is practically limited by the maximal TSC transmitted power (*P*_ST_) according to Equation (4), see Table 3.

The developed miniature millimeter-wave converters, described briefly in Section 7, are based on HMC1058 (Analog Devices) mixers, see [29]. Considering the +13 dBm assumed LO power, the maximum IF input power (up-converter) should not be, due to linearity requirements, higher than +3 dBm. Taking into account the 13 dB mixer conversion loss, the maximum up-converter output power reaches −10 dBm in front of the FIL1 and typically −13 dBm at the input of the ANT2 (Figure 3). So, a *G*_Smax_ equal to 0 dB and *P*_ST_ equal to −20 dBm can easily be ensured. Further, for proper target detection at a 3-m-range, an RSC equal to 1 m^2^ should be fully sufficient.

During system programming, it can also be useful to evaluate the topical RSC of the simulated target for the given TS setting. This is possible using Equations (3) and (4) (see Table 4) when *P*_ST_ = −20 dBm is considered. In near ranges, the values correspond well to the expected reduced RCS values. In far ranges, these achievable target RCSs are, with many dB to spare, sufficient for the simulation of all practical traffic scenarios.

A high probability of the accurate detection of targets by AR sensors is ensured by sufficient signal-to-noise ratios. The *SNR*_Rt_ values, defined by Equation (6), correspond to the AR sensor detecting real targets, and are referred to the input of its receiver. Calculated results are presented in Table 5. The noise bandwidth equals *B* = 1/*T*_sw_ = 12.5 kHz, which relates to standard FMCW radar chirp rates and signal processing. As a threshold for a reliable target detection, an *SNR*_Rtmin_ = 10 dB, defined in [6], was considered. This threshold corresponds to a target with an RCS equal to 20 m^2^ at a 100-m range, which corresponds well with the expected MRR capabilities.

The detection of targets simulated by the TS is affected by the additive RF noise generated by the TS and described by its noise figure (*F*_S_). The relation in Equation (16) enables the maximum *F*_S_ values (*F*_Smax_) to provide a *K*_RF_-times degradation of the *SNR*_RS_, with respect to the *SNR*_Rt_ values, to be calculated. Table 6 includes the calculated *F*_Smax_ values corresponding to a mere 1 dB signal-to-noise ratio drop (*K*_RF_ ≈ 0.794). Assuming the fully passive millimeter-wave down-converter structure, according to Figure 4 with the LNA and PA left out, and IF LNA at the input of the following IF circuits, the noise figure (*F*_S_) can easily attain approximately 20 dB (5 dB input filter loss, 13 dB conversion loss, 2 dB IF LNA noise figure). As can be deduced from Table 6, the allowed noise figures are, mostly, many tens of dB higher. Moreover, according to Table 5, the *SNR*_Rt_ values at low ranges are so high that even the concerned 1 dB drop cannot noticeably influence target detection. These parameters generally allow the TSs to be designed and realized in a relatively simple way by utilizing components with higher losses and noise figures, antennas with lower gains, etc.

As mentioned at the end of Section 4, when generating only a single target, the influence of the phase noise introduced by the TS LO, due to the correlation effects and integration in close vicinity to the beat frequency, is negligible. In the case of the TS enabling simulation of more targets at a single azimuth, the phase noise of the LO source within the TS can become more important. Equation (36) makes it possible to compute an LO’s phase noise pedestal value (${ℒ}_{\mathrm{Pb}}$), providing a *K*_PN_-times degradation of the *SNR*_Rt_, which is a signal-to-noise ratio corresponding to a real target in the defined ranges. Table 7 shows the results of Equation (36) for the allowed degradation of the *SNR*_Rt_ equal to 1 dB, i.e., *K*_PN_ = 10^−1/10^ ≈ 0.794. The evaluation assumes *f*_bc_ = 100 kHz, which corresponds to a mutual critical distance between two neighboring targets *R*_tc_ = 2 m. In the *R*_t_ = 3 m and 10 m lines, the condition *R*_t_ >> *R*_tc_ does not hold and certain deviations can be expected.

The interpretation of the calculated ${ℒ}_{\mathrm{Pb}}$ values presented in Table 7 is slightly more sensitive than the interpretation of the values of *F*_Smax_ included in Table 6. In the given 76–81 GHz frequency band, the common phase noise PSD pedestal values of ${ℒ}_{\mathrm{Pb}}$ of the available LOs range from −65 to −70 dBc/Hz. This explains why Table 7 indicates potential problems, especially in the 10 m and 3 m lines where relatively high PSD ratios are required. However, even at 30 m, the concerned values of the *SNR*_Rt_ are, according to Table 5, so high that a 1 dB drop cannot represent any substantial degradation of the sensor’s detection capabilities. What is more, the evaluated values of ${ℒ}_{\mathrm{Pb}}$ corresponding to the decisive *R*_t_ = 100 m range can easily be met, even by relatively “dirty” LOs.

A verification of the formulas derived for the analysis of additive RF noise and phase noise was performed using the Cadence AWR Visual System Simulator [30]. This solution was enforced by the fact that common AR sensors offer, at their digital busses, neither digitized internal signals nor any information on individual detected targets. These data are exclusively internal, while at the CAN busses, only the resulting instructions for the ECU (“start braking”, “turn to the left”, etc.) are usually available. Under such circumstances, computer simulations appeared to be the most feasible and reliable way of how to verify the developed analytical description.

The simulation schematic set for proper RF noise simulation, containing the AR sensor with parameters stated in Table 1, is shown in Figure 11. The generated FMCW radar chirp is noiseless and the first stage adding RF noise *kT*_0_*B* is the TS receiving antenna. Both mixers in TS are considered as noiseless and the overall TS noise figure *(F*_S_) is assigned to the A7 amplifier. This amplifier, with variable gain (*G*_S_), also ensures the simulation of an artificial target with an appropriate RCS in accordance with Equation (3). The influence of free-space losses between the AR sensor and TS are included in the AR sensor’s and TS’s receiving antennas. The time delay between the transmitted and the received signal, resulting in an appropriate beat frequency in the radar receiver, is ensured by RFDELAY blocks including free-space (FS) delays on the distance *(R*_S_) as well. The total noise contribution of the radar to the received signals is introduced by the A14 amplifier with a noise figure (*F*_R_) and 0 dB gain.

Figure 12 depicts the power spectrum of the beat signal present at the radar receiver’s output PORT1. The beat signal with a frequency of 1.5 MHz is caused by the simulation of an artificial target set to *R*_t_ = 30 m and an RCS *σ* = 10 m^2^. The *P*_Rt_ = −90.2 dBm power of the beat signal is in accordance with Equation (2), while the plotted noise floors correspond to the allowed *SNR*_Rt_ drops (*K*_RF_) equal to 0, 3 and 10 dB. To mimic the sampling of the beat signal during the duration of the chirp, the noise bandwidth of the simulation was set to *B* = 1/*T*_sw_ = 12.5 kHz. From Equation (12), the signal-to-noise ratios (*SNR*_RS_) at the radar receiver corresponding to the considered values of *K*_RF_ are equal to 27.8, 24.8 and 17.8 dB. Figure 12 also includes noise levels (*N*_RS_) calculated according to Equation (11), and the agreement with the simulated values is very good.

A validation of the phase noise correlation effects described in Section 4 was also performed using the VSS (Virtual System Simulator) schematic depicted in Figure 11. Small changes consisting of turning on the phase noise generation of the TS’s LO and turning off all other noise contributions were applied. The considered LO phase noise parameters are those from Figure 7. The resulting frequency plot of phase noise at the radar’s BB output (PORT1) is shown in Figure 13. The spikes repeating every 1/*T*_sw_ = 12.5 kHz are caused by periodic discontinuities in beat signals at the start of every frequency chirp. Aside from these spikes, very good agreement between Equation (27) and the VSS simulation is clearly observed.

To verify the developed analytical description presented in Section 5, a simulation of an LO phase noise influence when generating more targets at a single azimuth was also performed. The resulting spectrum of signals corresponding to the AR sensor input can be seen in Figure 14. The simulation corresponds to three artificial targets with an RCS σ = 100 m^2^ at distances 13, 15, and 17 m. The concerned targets produce beat signals with frequencies (*f*_b_) equal to 650, 750 and 850 kHz and powers (*P*_Rt_) equal to −65.6, −68.1 and −70.3 dBm. The phase noise parameters of the LO signal are shown in Figure 7. To produce a spectrum with higher resolution, the simulation was performed with a noise bandwidth equal to *B* = 1.25 kHz. To reduce side lobes, the Hann window was applied to the time domain signals before the FFT processing. In addition, to smooth the noise floor, 10 individual spectrum realizations were averaged. During the first simulation (marked as “noiseless”), all noise sources were turned off, whereas during the next simulation (“PN”), only the phase noise of an LO was introduced. The spectrum in Figure 14 should correspond to that in Figure 9, with the only difference being that the noise floor is about 10 dB less because of the ten-times narrower noise bandwidth used, as mentioned, to obtain a higher resolution. Considering this 10 dB shift, the agreement between Figure 9 and Figure 14 is very good.

## 7. Optimized HW Structure, System Programming

Results of the analyses presented in this article enable a significant simplification of the tested sensor simulator setup. Moreover, the definite derived formulas can also be claimed as a benefit for system programming.

Results obtained in Section 3 and Section 5 enable the structure depicted in Figure 4 to be simplified. Firstly, due to the high *F*_Smax_ values derived in Section 3 and evaluated in Section 6, the millimeter-wave LNA (MMIC chip, chip capacitors, non-trivial power supply, together with approximately 15 wire-bonds) can be eliminated. Secondly, taking into account the simulator output power (*P*_ST_) values derived in Section 2 and evaluated in Section 6, the power amplifier (PA) can also usually be omitted. The resulting down- and up-converters consist only of chip MMIC mixers and planar filters (see Figure 15).

The developed mmW converter includes a built-in 14 dB horn antenna and subharmonically pumped balanced mixer, and, together with an input RF filter reducing the mirror signals, IF filter and LO and IF connectors, occupies only an 80 × 40 × 8 mm volume. Identical converters can be used both for the RX and TX branches. A low price and low dimensions are extraordinarily important for the design and realization of the advanced simulator setups enabling a generation of a high number of targets at more azimuths using arrays of target simulating modules (see Figure 2). Small mmW units can be easily and densely placed in view of the tested sensor which enables complex radar pictures corresponding to the practical radar pictures seen by AR sensors in real traffic to be generated. Results of the phase noise analysis also enable the usage of relatively simple and therefore low-cost TS local oscillators.

The derived formulas are useful for system programming as well. In particular, for setting specific sensor simulator parameters enabling the generation and measurement of specific testing sequences and definite traffic scenarios. This concerns setting a suitable time delay, a Doppler frequency shift and the amplitude of a signal transmitted back from the TS to the AR sensor.

The amplitude of the signal transmitted by the simulator back to the AR sensor should be set according to Equation (4). If the gain (*G*_S_) is set according to Equation (3), the sensor detects the target at any range as a target with a constant RCS. This is an advantageous, but not necessarily mandatory, setting. In near ranges, from the radar sensor point of view, even slightly lower signal levels are still strong enough and ensure perfect detection. That is why, in this case, practically set values of *G*_S_ can be, without any difficulties, lower than those defined by Equation (3). Lower values of *G*s also mean a lower system gain is required and minor intermodulation problems, even in the case of *M*, in parallel simulated targets can occur. An effective RCS value, corresponding to this modified setting, can be evaluated using Equation (5) and the procedure can check if such an RCS still ensures a reliable target detection. The time delay of the *i*-th artificial target should be set according to
(37)$$\tau_{i}=\frac{2(R_{ti}-R_{S})}{c_{0}}-\tau_{L}$$
where *τ*_L_ describes the total latency of the simulator circuits, including the latency of both the analog and digital circuits. The Doppler frequency shift corresponding to a target’s radial velocity can be evaluated as
(38)$$f_{Di}=\frac{2f_{0}v_{i}}{c_{0}}\left({\vec{v}}_{0i}\cdot {\vec{\Psi }}_{0i}\right)$$

In this formula, *v_i_* describes the instantaneous velocity of the *i*-th simulated target, ${\vec{v}}_{0i}$ represents its unity velocity vector, while ${\vec{\Psi }}_{0i}$ stands for its unity azimuth vector defined by the AR sensor simulator’s TX antenna’s interconnecting line. In the digital domain, the Doppler frequency shift can be easily performed by multiplying the received and time-shifted signal by a complex exponential.

## 8. Conclusions

The employment of target simulators represents the most comprehensive way of how to test modern AR sensors. In addition to mere functionality checking, it also enables the evaluation of corresponding ECU reactions so crucial for passenger safety. Since advanced AR sensors offer high target resolution, the matching testing simulator should offer the simulation of a high number of targets at arbitrary ranges and azimuths, a feature still lacking in most available simulating devices.

This work indicates that these requirements cannot be met without a detailed theoretical analysis of the AR sensor simulator setup in question. This explains why the presented article includes a set of formulas derived for the description of all crucial system parameters. Especially detailed is the analysis of phase noise behavior indicating the range and LO quality dependences. For the design of the modern TSs, a description of the influences of this noise source for the generation of multiple targets can be important. All equations were verified by computer simulations using an AWR VSS, a professional CAD tool oriented to the analysis and design of RF systems. Agreements between calculated results and CAD simulations are, in all cases, very good.

Even relatively simple commercially available target simulators enabling the generation of targets at a single azimuth, with no azimuthal velocity component option, are exceptionally costly devices. HW simplification, enabled by the results of this work, can reduce the price of such sensor simulator setups by nearly an order and open the way for testing key automotive sensors, not only in the laboratories of automotive research departments but also in standard automotive testing practices.

The reduction in the complexity and price of the circuitry used in sensor simulator setups also opens the way to solutions based on arrays of a large number of TS transmitters offering a generation of high numbers of targets including movements in both radial and azimuthal directions. Such setups make it possible to generate targets radically close to practical radar pictures seen by AR sensors in a real operation. This should pave the way for enhanced quality in advanced automotive sensor testing.

## Figures and Tables

**Figure 1 sensors-20-02714-f001:**
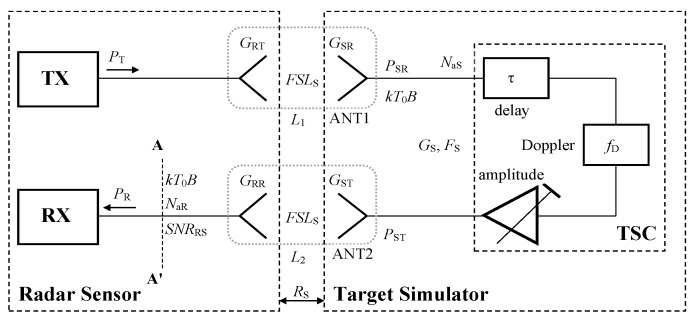
Block diagram with basic automotive radar (AR) sensor–target simulator (TS) relations.

**Figure 2 sensors-20-02714-f002:**
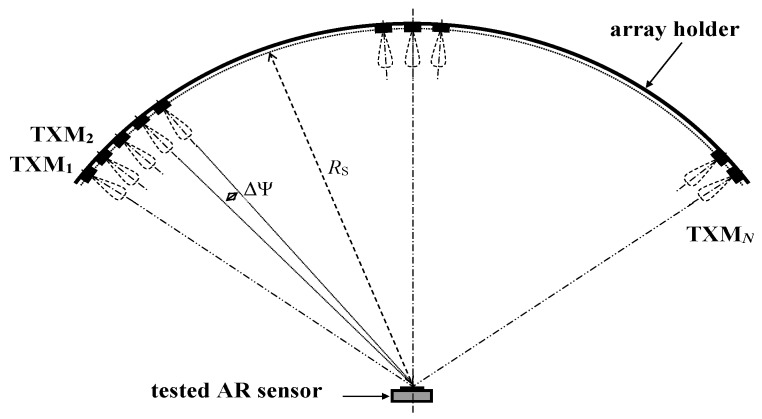
Testing setup based on an array of transmitting modules. The TS receiving antenna is situated anywhere in view of the AR sensor.

**Figure 3 sensors-20-02714-f003:**
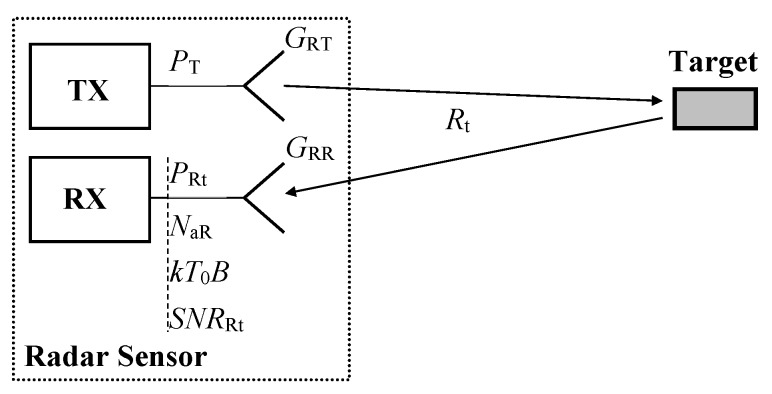
Sensor-target-sensor signal path with signal and noise parameters.

**Figure 4 sensors-20-02714-f004:**
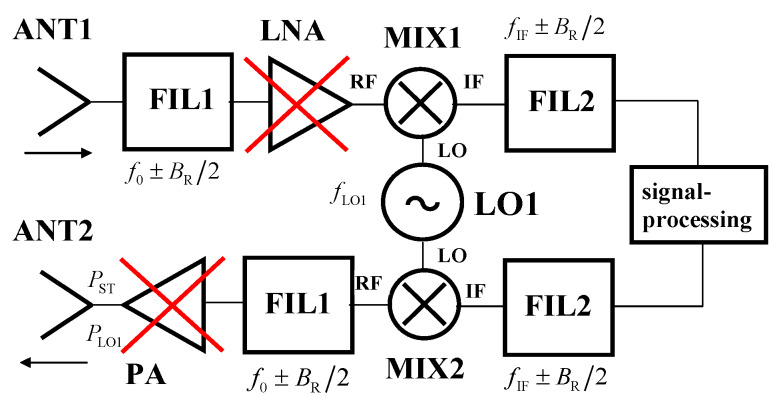
TS block diagram with common frequency converting circuits.

**Figure 5 sensors-20-02714-f005:**
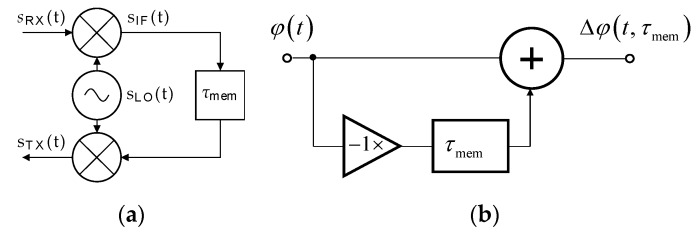
(**a**) Simplified schematic of TS used for the LO phase noise propagation analysis. (**b**) Equivalent circuit of the LO phase noise propagation.

**Figure 6 sensors-20-02714-f006:**
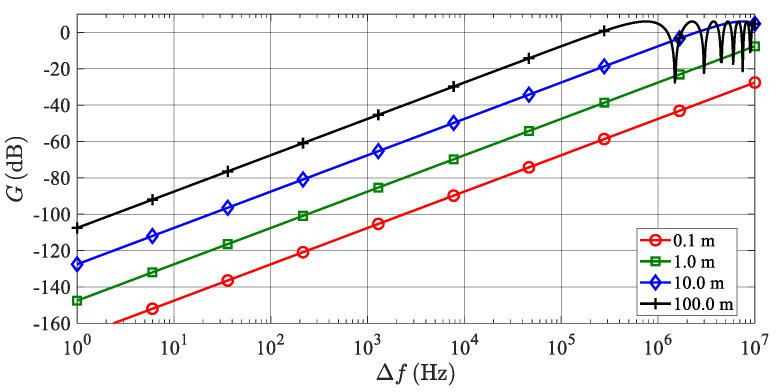
Transfer function of phase noise in TS and various target ranges. The function is plotted in the logarithmic frequency scale and positive values of Δ*f*.

**Figure 7 sensors-20-02714-f007:**
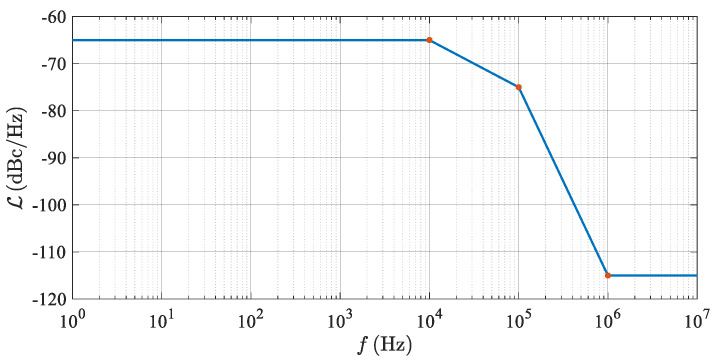
Phase noise spectral density ℒ(f) of a typical mmW LO source described as −65 dBc/Hz@10 kHz, −75 dBc/Hz@100 kHz and −115 dBc/Hz@1 MHz. The half-width of the phase noise pedestal is 10^5^ Hz.

**Figure 8 sensors-20-02714-f008:**
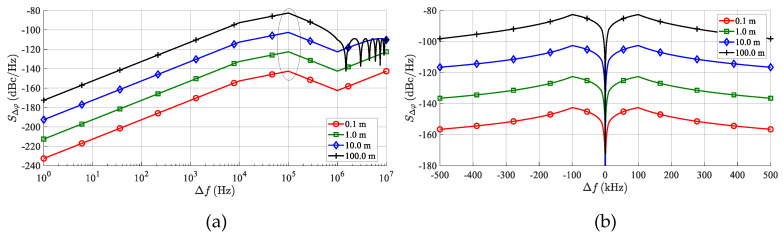
Phase noise spectral density (*S*_Δφ_ (Δ*f* )) of the TS up-converted output signal for an LO source with ℒ(f) shown in Figure 7. (**a**) Plotted for the positive values of Δ*f* in a log scale, frequency range 0–10 MHz. The marked frequency range is crucial when simulating more artificial targets at a single azimuth. (**b**) For both the positive and negative frequencies of Δ*f* in a linear frequency scale, at a frequency range of ±500 kHz.

**Figure 9 sensors-20-02714-f009:**
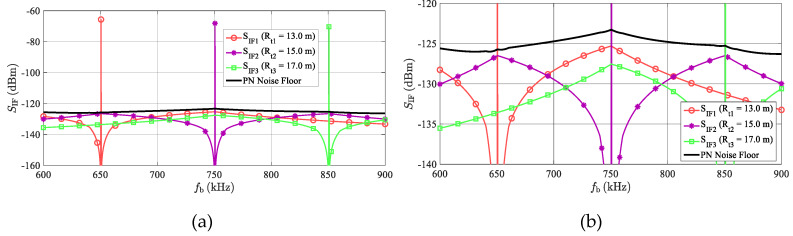
(**a**) MATLAB simulation of the power spectrum of received radar beat signals of three artificial targets at a single azimuth with a radar cross-sections (RCS) σ = 100 m^2^ and distances of 13, 15 and 17 m (mutual critical distance equals 2 m). Applied noise bandwidth equals *B* = 12.5 kHz and all simulated targets share the same LO characterized by the power spectrum density (PSD) of its phase noise defined in Figure 7. The frequency-modulated continuous-wave (FMCW) radar parameters considered are defined in Section 6. (**b**) Detail of the approximately calculated −125 dBm phase noise floor.

**Figure 10 sensors-20-02714-f010:**
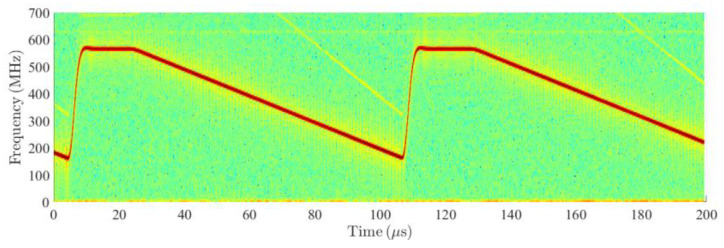
Measured spectrogram of received and down-converted signal from the MRR1 radar sensor with the external LO frequency set to 75.9 GHz. Transmitted chirps range from 76 to 76.5 GHz, while the active time of the chirp sweep is approximately 80 μs long.

**Figure 11 sensors-20-02714-f011:**
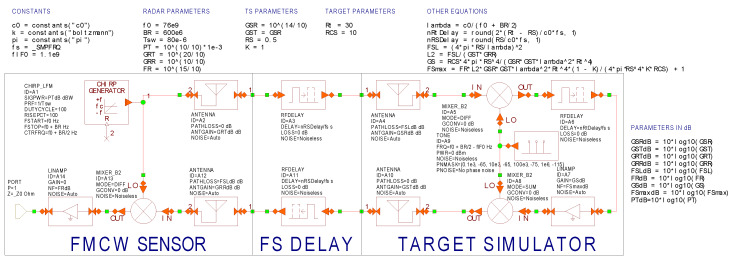
AWR Visual System Simulator schematic used for the analysis of the FMCW radar with the TS simulating a single artificial target. The radar model consists of the chirp generator A1, mixer A13 and transmitting and receiving antennas A2 and A12. The radar–TS round-trip delay is simulated by blocks A3 and A11. The TS model consists of receiving and transmitting antennas A4 and A10, and down- and up-converting mixers A5 and A8, with the shared local oscillator A9. Range of a single target is set by delay block A6 and its RCS by amplifier A7.

**Figure 12 sensors-20-02714-f012:**
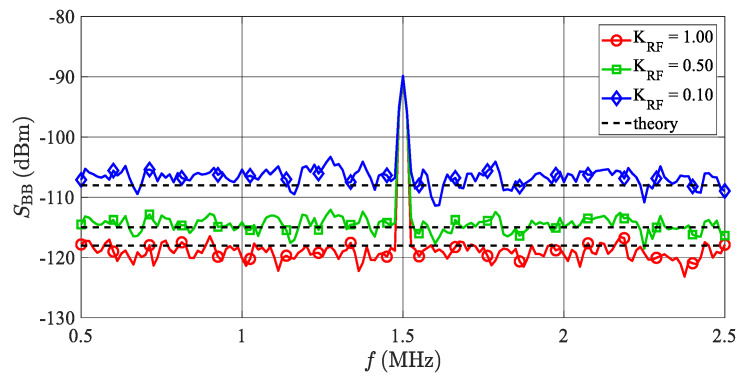
Simulated and calculated spectra of BB radar signals with additive RF noise. The beat signal at frequency 1.5 MHz is caused by an artificial target set to a 30-m range and an RCS equal to 10 m^2^. The amplitude of the beat signal is constant independently of the TS’s noise figure. Values of *K*_RF_ = 1, 0.5 and 0.1 correspond to TS’s noise figures *F*_S_ = 0, 85 and 95 dB, respectively. The “theory” lines represent the theoretical additive noise floors evaluated according to Equation (14).

**Figure 13 sensors-20-02714-f013:**
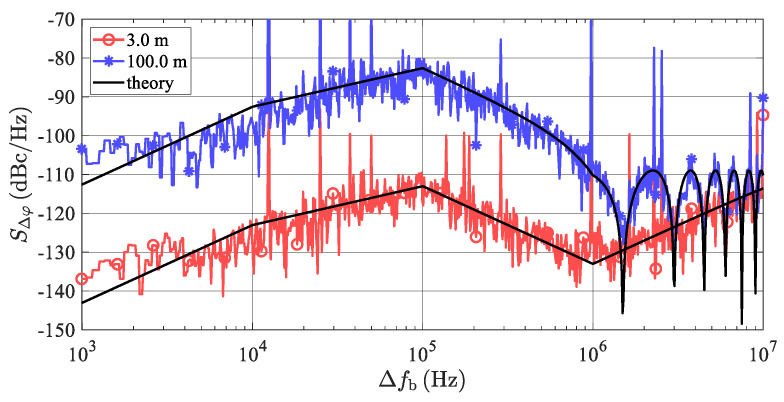
Simulation of phase noise at the radar BB output for targets at ranges of 3 and 100 m producing beat frequencies (*f*_b_) equal to 150 kHz and 5 MHz, respectively. The frequency axis of Δ*f*_b_ represents only the positive offset from the corresponding beat frequency. The spikes repeating every 1/Tsw = 12.5 kHz are caused by periodic discontinuities in beat signals at the start of every frequency chirp. The theoretical dependency of phase noise was computed using Equation (27).

**Figure 14 sensors-20-02714-f014:**
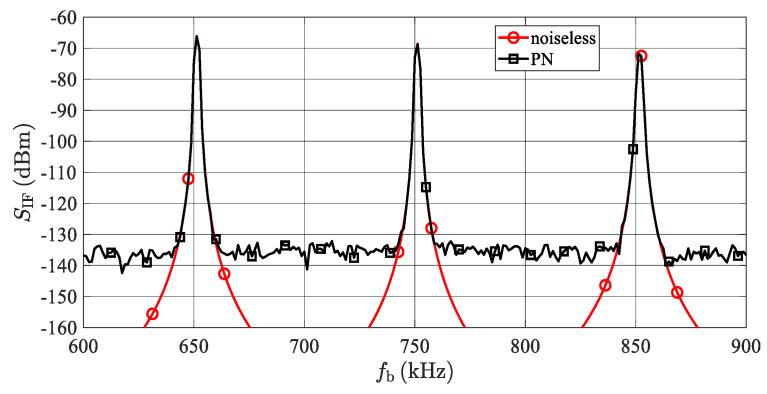
Simulation of beat signals at the AR sensor’s receiver coming from three targets at distances of 13, 15 and 17 m with an RCS 100 m^2^. In the “noiseless” case, the AR and TS are considered to be noiseless, and the only factors worsening the detection of targets is a main lobe of the Hann window utilized in the FFT processing. The “PN” case introduces phase noise to the LO signal in TS according to Figure 7. Thanks to the summation of noise contributions from both side targets, the center target is degraded, as predicted in Figure 9.

**Figure 15 sensors-20-02714-f015:**
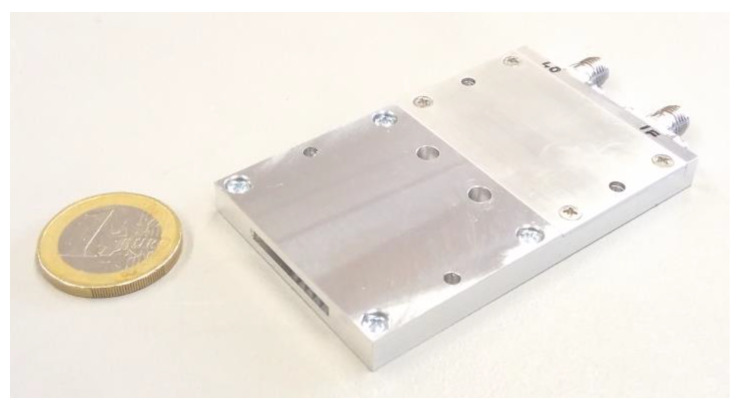
Fabricated mmW down- and up-converter containing a built-in 14 dB horn antenna, MMIC balanced mixer, planar filters and LO and IF connectors.

**Table 1 sensors-20-02714-t001:** Automotive medium-range radars (MRR) parameters.

MRR Parameter	Value
Start frequency of FMCW chirps	*f*_c_ =76 GHz
Bandwidth of FMCW chirps	*B*_R_ = 500 MHz
Sweep time FMCW chirps	*T*_sw_ = 80 µs
Radar transmitted power	*P*_T_ = 10 dBm
Radar transmitting antenna gain	*G*_RT_ = 20 dB
Radar receiving antenna gain	*G*_RR_ = 10 dB
Noise figure of radar receiver	*F*_R_ = 15 dB
Sampling frequency of IF signal	*f*_s_ = 10 MHz

**Table 2 sensors-20-02714-t002:** Calculated values of overall TS system gain (G_S_) (dB) necessary to simulate an artificial target with a constant RCS and specific ranges.

	σ	1 m^2^	10 m^2^	100 m^2^
*R* _t_				
3 m		0.0	10.0	20.0
10 m		−20.9	−10.9	−0.9
30 m		−40.0	−30.0	−20.0
100 m		−60.9	−50.9	−40.9

**Table 3 sensors-20-02714-t003:** Calculated values of the TS transmitted signal power (*P*_ST_) (dBm) necessary to simulate an artificial target with a constant RCS at specific ranges.

	σ	1 m^2^	10 m^2^	100 m^2^
*R* _t_				
3 m		−20.1	−10.1	−0.1
10 m		−41.0	−31.0	−21.0
30 m		−60.1	−50.1	−40.1
100 m		−81.0	−71.0	−61.0

**Table 4 sensors-20-02714-t004:** Calculated values of an achievable RCS of targets at various ranges with a limited maximal TS output power equal to –20 dBm.

*R*_t_ (m)	3	10	30	100
σ (m^2^)	1.02	126	10,200	1,260,000

**Table 5 sensors-20-02714-t005:** Calculated values of the signal-to-noise ratio (*SNR*_Rt_) (dB) at the AR sensor receiver input for various RCS values and ranges considering real targets.

	σ	1 m^2^	10 m^2^	100 m^2^
*R* _t_				
3 m		57.8	67.8	77.8
10 m		36.9	46.9	56.9
30 m		17.8	27.8	37.8
100 m		−3.1	6.9	16.9

**Table 6 sensors-20-02714-t006:** Calculated values of maximal noise figures (*F*_Smax_) (dB) of TS circuits causing the degradation of the *SNR*_Rt_ by 1 dB.

	σ	1 m^2^	10 m^2^	100 m^2^
*R* _t_				
3 m		49.2	39.2	29.2
10 m		70.1	60.1	50.1
30 m		89.2	79.2	69.2
100 m		110.1	100.1	90.1

**Table 7 sensors-20-02714-t007:** Calculated values of the LO phase noise pedestal (${ℒ}_{\mathrm{Pb}}$) (dBc/Hz) causing degradation of the *SNR*_Rt_ by 1 dB; the degradation relates to the middle target when generating three targets with identical RCSs and a critical *R*_tc_ = 2 m distance.

	σ	1 m^2^	10 m^2^	100 m^2^
*R* _t_				
3 m		−68.9	−78.9	−88.9
10 m		−59.0	−69.0	−79.0
30 m		−49.6	−59.6	−69.6
100 m		−39.1	−49.1	−59.1

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
