# Peer review of "On the Testing of Advanced Automotive Radar Sensors by Means of Target Simulators"

_sensors, 2020, doi:10.3390/s20092714_

Round 1

Reviewer 1 Report

it is required to improve the detail of each source of literature. Describe for each source individually. In the article, you use the link in the section "Introduction" immediately to 3 sources.

driving concepts [1–3].

Information on a variety of different TS types can be found, e.g., in [10–18]. 

Mathematical structures and formulas are enough. the drawings fully reflect the structure.
Add detailed chart descriptions Figure 10-13

Add digital metrics to 8. Conclusions

Reviewer 2 Report

It was a pleasure reviewing your paper. I suggest a few modifications:

1. As far as I understood, the main contribution is about phase noise (for single- and multi- target cases). Contents in Section 2 and 3 are quite trivial once the system model is established. The title and abstract should focus on your phase noise analysis for radar target simulators.

2. Overall, contents in Section 2 and 3 should be shortened. Contents in Section 4 and 5 should be explained in more detail.

3. In Section 2, since surface power densities are not used later, equations 1 and 3 seem redundant.

4. Section 4 Figure 5 is very important for your phase noise model, so it should be explained in detail, such as the exact component each equivalent circuit element represents. Tau_mem could change time to time in real hardware, or there could be mulitple paths with different tau_mem's for phase noise to propagate. The paper should explain the reasoning behind the equivalent circuit.

5. Based on the (new) detailed explanation for the new circuit element, equations like 25,26 should be derived from models for LO, Mixers, and filters.

6. Line 316: The paper should explain why the total phase noise is the sum of phase noise for each target.

7. Section 6. Line 385 Obtaining parameters that are unavailable in public documents can be tough for other researchers, too. The paper should show the experiment setup, and the estimation methods for important parameters.

8. Figure 11, 12: The text should explain what equation the "theory" line represents.

9. Figure 11: The equation behind "theory" line should include the artificial target signal.

10. Section 7: If possible, Measurements with the fabricated equipment should be presented.

11. The conclusion should focus about phase noise.

Reviewer 3 Report

  • throughout your manuscript, please first give the respective description and then the corresponding symbol; i.e. instead of “The S_SR surface power density” please use “The surface power density S_SR”.

  • in eq. (11) and beyond, the definition of SNR/bandwidth is confusing; for 11 you define the bandwidth B to be dependent on the signal processing applied; i.e. you obviously define the SNR in the 2D Frequency domain, including the coherent Gain of the 2D FFT; why not simply define the SNR at the stage of ADC data? Then B would clearly correspond to the IF bandwidth (including analog receiver + CIC decimation stages) used in the radar system; thus the SNR definition would be more decoupled from the signal processing

  • In particular the definition of bandwidth B from eq. (12) is not defined in 2D frequency domain (since there is no 2D FFT processing in the AR)

  • your manuscript is based on simulations only; in the final section however, you include a design which makes use of the simplifications (no LNAs) which you derived from your design formulas; however, to present no measurement of that design; you should however include measurements of that design and carefully compare those measurement to your simulations to clearly prove that your design methodology is valid

  • your article needs major revision regarding English language and grammar

Round 2

Reviewer 3 Report

Thank you for improving your article with this revision. The quality of your submission has greatly improved from your additions and corrections.